# Transmission of Vaccination Attitudes and Uptake Based on Social Contagion Theory: A Scoping Review

**DOI:** 10.3390/vaccines9060607

**Published:** 2021-06-05

**Authors:** Pinelopi Konstantinou, Katerina Georgiou, Navin Kumar, Maria Kyprianidou, Christos Nicolaides, Maria Karekla, Angelos P. Kassianos

**Affiliations:** 1Department of Psychology, University of Cyprus, Nicosia 1678, Cyprus; pconst09@ucy.ac.cy (P.K.); katerinageorgiou1588@gmail.com (K.G.); mkypri11@ucy.ac.cy (M.K.); mkarekla@ucy.ac.cy (M.K.); 2Yale Institute for Network Science, Yale University, New Haven, CT 06520, USA; navin.kumar@yale.edu; 3Department of Business and Public Administration, University of Cyprus, Nicosia 1678, Cyprus; nicolaides.christos@ucy.ac.cy; 4Initiative on the Digital Economy, MIT Sloan School of Management, Cambridge, MA 02142, USA; 5Department of Applied Health Research, UCL, London WC1E 6BT, UK

**Keywords:** vaccination, immunization, vaccine hesitancy, social contagion theory, social network analysis, scoping review

## Abstract

Vaccine hesitancy is a complex health problem, with various factors involved including the influence of an individual’s network. According to the Social Contagion Theory, attitudes and behaviours of an individual can be contagious to others in their social networks. This scoping review aims to collate evidence on how attitudes and vaccination uptake are spread within social networks. Databases of PubMed, PsycINFO, Embase, and Scopus were searched with the full text of 24 studies being screened. A narrative synthesis approach was used to collate the evidence and interpret findings. Eleven cross-sectional studies were included. Participants held more positive vaccination attitudes and greater likelihood to get vaccinated or vaccinate their child when they were frequently exposed to positive attitudes and frequently discussing vaccinations with family and friends. We also observed that vaccination uptake was decreased when family and friends were hesitant to take the vaccine. Homophily—the tendency of similar individuals to be connected in a social network—was identified as a significant factor that drives the results, especially with respect to race and ethnicity. This review highlights the key role that social networks play in shaping attitudes and vaccination uptake. Public health authorities should tailor interventions and involve family and friends to result in greater vaccination uptake.

## 1. Introduction

Vaccination attitudes and uptake can spread within networks and influenced by each individual’s social contacts [1]. According to the Social Contagion Theory, an individual can exhibit behaviour modelled by another person or adopt the attitudes of members of their social network [2,3,4]. For example, the food choices of one spouse can predict similar food choices of the other spouse [5], and having an obese spouse can predict by up to 40% whether the other spouse will become obese [6]. Therefore, the Social Contagion Theory can inform our understanding on how one’s health outcomes can be influenced by their social network and how attitudes and behaviours are transmitted from one individual to another [7]. This can be translated to vaccination research and policy where understanding how vaccination attitudes and uptake are spread within social networks can inform public health policies and interventions to improve vaccination rates.

A number of network topological features are involved in social transmission of attitudes and behaviours within a network including social ties (i.e., the relationship between individuals such as friendships) and the quality of the relationships [2]. Further, social transmission can be influenced by the position of a person within a network such as the person’s centrality, which may influence attitudes and behaviours to a greater extent than those who are in the periphery of the network [8]. For example, in one study [9], adolescents who were more centrally located in the network of friends and siblings were more influential upon other adolescents’ drug use and sleep outcomes than those who were not at the core of the network. Another topological feature influencing social transmission consists of clustering between individual behaviours in a social network (i.e., co-occurrence of a trait of interest among network members) which is quite prevalent across physical exercise, happiness and obesity [2]. Clustering might occur as a compendium of multiple reasons including: (a) homophily of preferences which refers to the tendency of similar individuals to connect with each other [4,10,11], (b) social influence whereby social network members might exert causal social influence on the attitudes and behaviours of the individual [2,11], (c) confounding factors which refers to the propensity that certain areas of a social network are subject to same externalities [4], and (d) simultaneity which refers to the tendency for connected individuals in a social network to co-influence each other [4].

The way that attitudes and behaviours are spread have been examined in both egocentric and sociocentric networks. Egocentric refer to networks of individuals that are mapped with information provided on their ties and sociocentric are networks that entail the interactions of all members of a community or group [7]. Specifically, there have been studies examining the spread of happiness [11], food choices [5], obesity [6], smoking [12], depression [8], alcohol consumption [13], and most recently of social distancing behaviours during the COVID-19 pandemic [14] in several social networks. A significant effect of social networks in individuals’ attitudes and behaviours was identified in all studies [5,6,8,11,12,13,14]. For example, in a longitudinal sociocentric study [12], a sibling, friend or spouse who stopped smoking influenced the decrease in an individual’s smoking by 25–67%. 

Vaccination attitudes and uptake may also spread within social networks. A decision to vaccinate or not is usually made based on local vaccine policies, information from social media, as well as an individual’s social network [15,16]. The rates of under-vaccinated adults and children are increasing and this can be attributed to vaccination hesitancy [15,16], which refers to the delay in accepting or refusing vaccination despite its availability [16]. Vaccination hesitancy is an important and complex problem that contributes to outbreaks of diseases and to increased mortality rates [15,16,17]. Examining the influence of social networks in individuals’ vaccination attitudes and uptake is particularly of importance given that at present the world is in the midst of a pandemic for which vaccines are produced and appear to be the only solution to manage the COVID-19 pandemic. For vaccination programs to be successful, a critical mass of the population needs to receive the vaccine, thus the spread of vaccination hesitancy is a major barrier that governments are facing globally. 

This scoping review aims to collate evidence from the literature on how vaccination attitudes and uptake are spread from one individual to another in sociocentric and egocentric networks. The main objectives are: (a) to describe the features of social network membership within the included studies, and (b) to examine the evidence from the included studies on how vaccination attitudes and uptake of individuals are influenced by their social networks. 

## 2. Materials and Methods

This scoping review was conducted at the midst of the COVID-19 pandemic in 2020 when vaccines were produced and distributed. Thus, rapid evidence was needed in order to inform policies on tackling vaccination hesitancy and provide guidance on rolling out national vaccination campaigns effectively. This scoping review was registered with PROSPERO (registration number: CRD42020219300) and followed the PRISMA guidelines for reporting scoping reviews [18]. Data supporting the findings of this study are available in Open Science Framework (OSF) (osf.io/5 gucf). 

### 2.1. Eligibility Criteria

Peer-reviewed studies and grey literature (e.g., dissertations) were eligible for selection. The PICO method was used to determine the inclusion criteria for this review [19]: (a) P (Participants): No inclusion criteria were set for the demographic characteristics of participants, (b) I (Intervention): Not applicable, (c) C (Comparison): Not applicable, and (d) O (Outcome): Evidence on the influence of social networks on vaccination attitudes and uptake for any type of vaccination. Eligible studies needed to be longitudinal, observational (including cross-sectional, prospective, and retrospective), qualitative, or randomized controlled trials and published in English language. Studies were excluded if: (a) published in a language other than English; and (b) were letters, reviews, editorials, conference abstracts, or case studies.

### 2.2. Search Strategy and Study Selection

Relevant studies (no date restrictions applied) were identified by searching the electronic databases of PubMed, PsycINFO, EMBASE, and Scopus. Searches were conducted until December 2020. A defined search strategy was undertaken using the following terms based on title and abstract: “social contagion” or “social network”, combined with the terms “vaccine”, or “vaccinate” or “vaccination” or “anti-vaccination” or “immunization”. The full search strategy is available in Appendix A.

Articles were screened for eligibility at title/abstract and full-text screening stages, by two authors independently (PK, KG). Inter-rater reliability (IRR) referring to the extent to which the two screeners agreed, was calculated using the percent agreement (number of agreement scores divided by the total number of scores) and Cohen’s kappa (a more robust measure for IRR) [20]. An almost perfect agreement was observed between the two screeners during title/abstract screening (IRR = 94%; k = 0.86) and for the full-text screening stage (IRR = 96%; k = 0.86). Any discrepancies were resolved in research team consensus meetings.

### 2.3. Data Extraction and Synthesis

A data charting form was used to extract the data (see Appendix B). From all included studies, characteristics of the study and population, analytical approach (e.g., social network analysis), description of type of network (e.g., egocentric), and main findings (e.g., centrality, spread) were extracted. A narrative synthesis approach [21,22] was used to analyze, summarize and interpret findings of included studies. This narrative synthesis described the studies and participants’ characteristics and collated the evidence on how vaccination attitudes and uptake are spread within social networks. 

## 3. Results

### 3.1. Study Characteristics

A total of 1043 studies were identified. After removing duplicates and screening the titles, the full text of 24 studies were screened and 11 retained (see Figure 1). 

The characteristics of the 11 included studies are presented in Table 1. These were published between 2011 and 2020, with the majority conducted in the USA (*n* = 7, 63.6%). All studies were cross-sectional. The studies examined self-vaccination (*n* = 7, 63.6%) and childhood vaccinations (*n* = 4, 36.4%). Self-vaccination types included human papillomavirus (HPV) (*n* = 3, 27.3%), seasonal flu (*n* = 2, 18.2%), influenza H1N1 (*n* = 1, 9.1%), and both influenza and seasonal flu (*n* = 1, 9.1%). Parents’ attitudes on vaccinations and uptake were examined for all routine childhood vaccinations (e.g., measles; *n* = 3, 27.3%) and HPV (*n* = 1, 9.1%). Populations varied including university students (*n* = 4, 36.4%), parents of children aged less than 18 months (*n* = 2, 18.2%) and 10–12 years (*n* = 1, 9.0%), pregnant women with first child (*n* = 1, 9.0%), females aged 18–65 (*n* = 1, 9.0%), adults working together in organizational settings (*n* = 1, 9.0%), and children up to 12 years (*n* = 1, 9.0%). 

### 3.2. Description of Methodology and Analytical Approach

Most of the included studies used social network analysis to examine influence of social networks on vaccination attitudes and uptake (*n* = 7, 63.6%) and with the remaining using logistic regression models (*n* = 4, 36.4%). Convenience sampling methodology (*n* = 10, 91.0%) was mostly used followed by stratified sampling (*n* = 1, 9.0%). Most studies collected data using online or paper-based questionnaires (*n* = 8, 72.7%) and interviews (*n* = 3, 27.3%). All studies used egocentric networks to examine the outcomes of vaccination attitudes and uptake. Findings of each study are outlined in Table 2.

### 3.3. Transmission of Vaccination Attitudes and Uptake within Social Networks 

Across studies, vaccination attitudes and uptake of participants were highly influenced by their social networks (see Figure 2 for a summary). Positive attitudes on self and childhood vaccinations were influenced by social networks’ positive attitudes [24,25,27,28,29,30,32,33], whereas having vaccinated people in networks was related to increased likelihood of participants to be vaccinated [26] or vaccinate their child [29]. Similarly, negative attitudes and lower vaccination uptake were influenced by social networks’ negative attitudes and lower uptake [23,28,32,33]. Positive attitudes referred to beliefs that childhood vaccines are effective at protecting children, reduce the risk for developing a health condition (e.g., cancer), and are safe and effective [24,25,26,27,28,29,30,32,33]. Negative attitudes referred to beliefs that vaccines are dangerous or unsafe, might cause symptoms and are in an experimental stage [28,32,33].

Most of the included studies (*n* = 8, 72.7%) reported that family and friends/peers significantly influenced self and childhood vaccination attitudes and uptake. In contrast, only two studies (18.2%) reported that healthcare providers [29,32] and co-workers [23,27] and one study (9.0%) that politicians [29] significantly influenced vaccination attitudes and uptake. For example, Casillas et al. [24] reported that discussing about the vaccine with family and/or friends significantly increased the odds for perceiving the HPV vaccine as effective (Odds Ratio = 1.98, 95% CI: 1.04–3.78) compared to discussing them with the healthcare provider which had a non-significant effect (Odds Ratio = 1.71, 95% CI: 0.86–3.39) Some studies [24,25,28,29,32,33] found that participants held more positive attitudes towards self and childhood vaccinations when they were discussing them with family and friends/peers who held similar attitudes, or when they perceived their family and friends/peers holding positive attitudes towards self and childhood vaccinations. Vaccination uptake for self or children increased when the individuals’ network was comprised mostly by vaccinated family and friends [28,31,32] or when parents observed their peers vaccinating their child [29]. Conversely, vaccination uptake for self or children decreased if family and friends were vaccine hesitant or held negative attitudes toward vaccinations [28,33]. Moreover, in a sample of foundation doctors, participants were more likely to get vaccinated when they had a higher number of vaccinated neighbours in their network [26]. Additionally, in a sample of individuals working together in organizations (e.g., health and social services, financial services), they were more likely to get vaccinated when they perceived their co-workers holding positive attitudes towards vaccinations [27]. 

Regarding mechanisms underlying transmission within networks, frequency of communication between network members and prolonged exposure to positive (e.g., safety, effectiveness) or negative (e.g., dangerous, ineffectiveness) self and childhood vaccination attitudes explained transmission in social networks. Specifically, participants held more positive attitudes towards self and childhood vaccinations and greater likelihood to get vaccinated or vaccinate their child when they were more frequently exposed to positive vaccination attitudes than negative [28,32]. In addition, participants were more likely to vaccinate their child when they frequently discussed vaccinations with family and friends who held positive vaccination attitudes [29]. Self-vaccination also increased when participants felt that their significant others wanted them to be vaccinated or when they wanted to comply with the vaccination behaviour of their social networks [27,33].

Clustering of attitudes was identified in a sample of co-workers, with participants tending to share similar vaccination attitudes with people working within the same group [27]. Participants were more likely to get vaccinated when people working within the same group were vaccinated or when they perceived them as supporters of vaccinations. In contrast, no clustering was identified in university students, with vaccinated students being as likely as non-vaccinated students to be friends [25]. Centrality evidence was only reported by one study [29], in which it was found that the centrality of peers and opinion leaders (i.e., political, religious and traditional medicine providers) within social networks did not influence mothers’ behaviour to vaccinate their children. 

Further, homophily was found to influence the transmission of vaccination attitudes and uptake within social networks [28,29,30,31]. Out of the five (45.5%) studies that reported results on homophily, four (80.0%) observed the presence of homophily in the social network, with race/ethnicity reported by all studies influencing the formation of networks [28,29,30,31]. Additionally, members of social networks presented with similarities in educational level, and parental and marital status [28,29,30,31]. For example, Goldberg [29] and Fu et al. [28] identified that peers who influenced parents’ decision to vaccinate their children were more likely to be of the same race/ethnicity (African Americans, Muslims, Hausa), gender (females), marital status (married), be parents, and with similar educational level (no formal education). Furthermore, Mascia et al. [31] found that vaccinated children tended to have other vaccinated children in their networks with similar ethnicity and class. Hernandez et al. [30], found that pregnant women with their first child tended to have a social network with similar education, with well-educated women having a well-educated network supporting vaccination uptake. Therefore, individuals tend to have homogeneous networks (see Figure 2 for a summary of the mechanisms). Suggestions for further research based on the type of network, vaccination and attitude are presented in Table 3. 

## 4. Discussion

Eleven studies were identified in this review examining how self and childhood vaccination attitudes and uptake are spread within social networks. Our results suggest that social networks play an important role in shaping positive and negative attitudes and in vaccination uptake. Individuals held more positive attitudes and had a greater likelihood to either self-vaccinate or vaccinate their children if their network was mostly comprised by people holding positive attitudes (e.g., vaccination safety and effectiveness), were vaccinated, or were perceived as vaccine supporters. Frequent discussion on vaccinations with family and friends/peers who held positive attitudes or were vaccinated, and higher exposure to positive attitudes also increased the likelihood of vaccination uptake. In the same way, negative attitudes and lower vaccination uptake were transmitted within networks. Since all people are connected to other people, the effects of an intervention which is delivered to an individual might be indirectly diffused to their social network [34,35]. Clinicians and policymakers could consider network structure of for example communities and general practice patients, in order to result in higher diffusion of interventions’ effect.

It is important to note that by simply being exposed to or discussing vaccinations with others does not imply that an individual will adopt the same behaviour [36]. Social transmission is a complex process involving an individual’s knowledge, skills, motivation and attitudes, and opportunities provided by their network [1]. For example, according to the COM-B model [36], a behaviour change may occur when an individual has opportunities to enable the behaviour such as positive support from family and friends together with other attributes such as the psychological and physical capacity, capabilities and motivation to perform the behaviour. In addition, the Theory of Planned Behaviour (TPB) [37] suggests that the behavioural intentions for performing a behaviour are shaped by the beliefs of significant others and motivation to comply with them, positive or negative attitudes, and perceived behavioural control over the desired behaviour. Even if a person perceives the vaccine as effective and is available to them, if social network members do not perceive it as effective or are not vaccinated, vaccination hesitancy is more likely to occur [38]. High applicability of the COM-B and TPB concepts is observed in our review, as social influence and motivation to comply with the behaviour of significant others were evidenced, with some of the included studies reporting that vaccination uptake increased when participants wanted to comply with the vaccination behaviour of the network [27,33]. Individuals may also adopt the vaccination attitudes of their social network or get vaccinated as a result of social norms; to fit in or to be socially accepted [39,40,41]. Therefore, vaccination uptake should be understood as an interplay of factors involving not only the individual but also his social network.

Family and friends/peers appeared to have more influence on individuals’ attitudes and vaccination uptake than other members of social networks such as healthcare providers and neighbours. This is not uncommon among health outcomes as obesity has been found to be transmitted in a greater extent from those in the immediate environment of the person, siblings and spouses compared to neighbours [6]. The quality of the relationship and the frequency of communication with network members might be more critical in social transmission than the expertise, authority and knowledge of other network members; yet these have not been examined in relation to vaccine behaviours. Future studies can examine the factors underlying how family and friends/peers influence vaccination attitudes and uptake compared to other network members. In addition, clinicians and policymakers are recommended to include social network members in interventions or provide educational family-based programs on vaccinations. For other health behaviours such as smoking-cessation, programs that include peer support are more effective than those who do not involve social network members [35,42]. Further, including network members may result in greater diffusion of an intervention’s effects within networks than individual-based approaches as individuals tend to benefit from indirect exposure to an intervention [34].

Substantial homophily was identified in included studies, with race/ethnicity playing the most important role in forming social networks. Other factors identified being similar between network members were education level, parental and marital status. Existence of homophily within networks is a methodological challenge for researchers as it results into homogeneous samples with restrictions of including people from various backgrounds and thus possibly confound estimates of effects of social networks [43,44]. One way to overcome homophily is by conducting longitudinal studies in order to examine social networks dynamically over time [10]. Additionally, interventions or educational vaccination campaigns could be tailored to the target populations especially with ethnic minorities, who can hold specific beliefs and barriers to vaccination uptake and may not be influenced by individuals of other ethnic background. Tailored interventions are preferable by individuals, can be associated with better health outcomes and present with higher adherence [45,46,47]. For example, in parental populations, public health vaccination campaigns could emphasize the protection of their children from health conditions, whereas in non-parental populations could emphasize the protection of themselves and significant others. Targeting each network using recommendations for campaign messaging, such as the use of short, risk-reducing or relative risk framing messages with clear and simple language [46], could possibly reduce vaccine hesitancy. Furthermore, our evidence suggests that identifying and intervening to networks with predominantly negative attitudes towards vaccinations can also protect individuals in the network who hold neutral or positive attitudes.

Future studies can conduct longitudinal experimental research to better understand the mechanism of spread of vaccination attitudes and uptake, infer causal relationships, and determine how social networks are formed and function. In addition, although clustering was identified in one study [27], the mechanisms underlying clustering could not be understood as the research design was cross-sectional. Possible explanations of clustering might be due to homophily as individuals might have chosen to cluster with co-workers with similar vaccination attitudes, or induction as the members of the group might have exerted social influence on the individual [13]. In contrast, no clustering was identified within a medical student network [25], with vaccinated students being as likely as non-vaccinated students to be friends, possibly due to the way people make friends in younger ages as opposed to how they form or maintain relationships in older ages. In younger ages people tend to have a higher number of friendship networks with emphasis given on their common interests, compared to older ages with more emphasis given in mutual beliefs [48]. Future studies are advised to examine mechanisms underlying clustering. Additionally, future studies are suggested to examine the impact of specific sociodemographic characteristics such as age and gender in forming social networks and in the spread of vaccination attitudes and uptake as either were not examined in the included studies or mixed findings were observed (i.e., same vs. opposite gender) [28,31]. Based on promising findings of previous studies on the high impact of centrality in health behaviours such as depression [11], future studies are also advised to examine centrality in social networks and its influence on transmission of vaccination attitudes and uptake. Further, examining the influence of social networks using sociocentric networks is needed as all included studies used egocentric networks. Sociocentric networks may offer the opportunity for more robust evidence of contagion in entire networks as information are collected from both the individual and their network members [7]. Researchers interested in examining transmission of vaccination attitudes and uptake within social networks should additionally refer to Table 3 for specific recommendations for each type of network (e.g., family), vaccinations (e.g., HPV) and attitudes (positive vs. negative).

### Limitations

As this study was a rapid scoping review, quality assessment of included studies was not conducted. Furthermore, the studies included a variety of populations (e.g., students, mothers, parents, children), making it more complex to assess or synthesize all studies under the same rubric. In addition, although we searched several databases, we may have missed some studies due to the inclusion of studies published only in English.

## 5. Conclusions

Vaccination attitudes and uptake can be highly influenced by ones’ social network. Being exposed to positive attitudes, frequently discussing vaccinations with family and friends/peers or wanting to comply with their behaviour increases the likelihood of an individual to get vaccinated or vaccinate their child. Homophily was observed within networks with individuals tending to have similar networks, especially in respect to race and ethnicity. Public health authorities and policymakers could consider including social networks of individuals when delivering interventions or educational campaigns on vaccinations to benefit members of the network who can be influenced negatively towards vaccinations. Tailoring interventions and campaigns to the target populations is strongly advised. Only then may vaccine hesitancy rates be reduced, contributing to decreased mortality rates and better health outcomes, especially during epidemic outbreaks [15,16,17].

## Figures and Tables

**Figure 1 vaccines-09-00607-f001:**
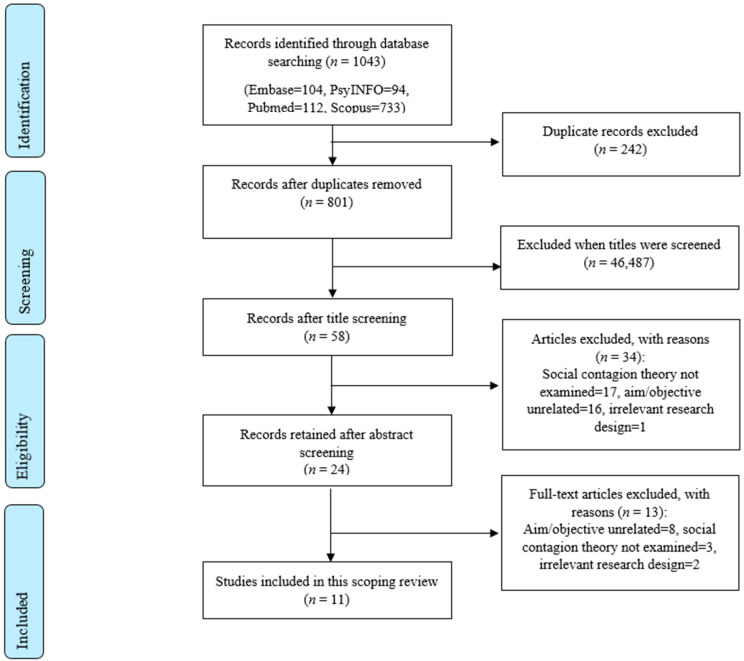
Flow diagram of information detailing the database searches, the number of titles and abstracts screened and excluded, and the full texts retrieved and excluded.

**Figure 2 vaccines-09-00607-f002:**
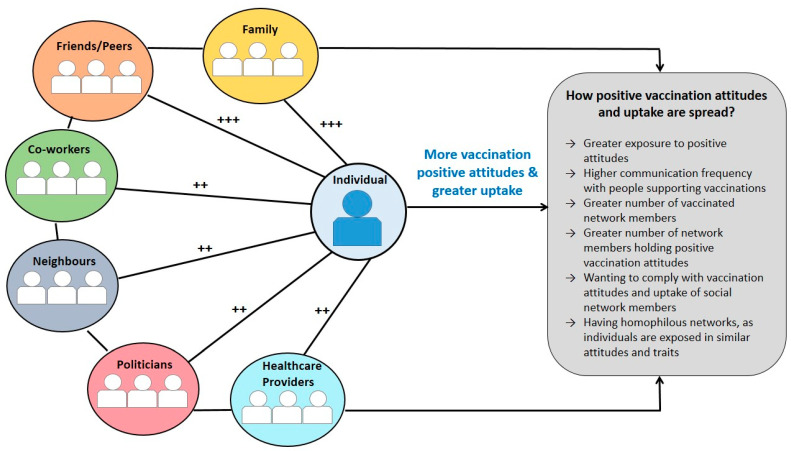
A summary of findings explaining how vaccination attitudes and uptake are transmitted within social networks. *Note.* ++ Lower influence on vaccination attitudes and uptake of individuals compared to other network members (family, peers and friends) based on the total number of studies reporting this information; +++ Higher influence on attitudes and vaccination uptake of individuals compared to other network members (neighbours, co-workers, politicians, healthcare providers) based on the total number of studies reporting this information.

**Table 1 vaccines-09-00607-t001:** Descriptive information of included studies (*n* = 11).

Study ^1^	Country	Aim	Population	Sample Size	% Females	Age ^2^ (M, SD)	Education ^3^	Vaccination Type
Brunson (2013) [23]	USA	To examine the effect of parent, people and source networks on parents’ vaccination decisions.	First-time parents with children aged ≤18 months	196	92.3%	31.3 (4.7)	Bachelor’s degree: 46.4%	Childhood vaccinations (type not specified)
Casillas et al. (2011) [24]	USA	To examine the influence of hearing or discussing the vaccine with family/friends on perceived HPV vaccine effectiveness.	Low-income, minority women aged 18–65	294	100%	43.9 (0.3)	Highest education—high school: 40.2%	HPV
Edge et al. (2015) [25]	UK	To examine the effects of social networks on influenza vaccination decision.	Primary undergraduate medical students at Lancaster Medical School	217	NR	NR	Primary undergraduate medical degree: 100%	Seasonal flu
Edge et al. (2019) [26]	UK	To evaluate the effect of social network on seasonal influenza vaccination uptake by healthcare workers.	Early career doctors working at the Pennine Acute Hospitals NHS Trust	138	49.3%	NR	Early career doctors at year 1: 72.4%	Seasonal flu
Frank (2011) [27]	USA	To explore how social norms about health are understood in adults working together in organizational settings.	Adults who work together for the same organization in the same physical location	152	57.0%	30–49: 49.0%	4-year college degree: 41.0%	H1N1
Fu et al. (2019) [28]	USA	To examine the influence of social networks for HPV vaccine among African American parents.	African American parents of children aged 10–12 years	353	94.1%	Median: 37 (NR)	≤High school graduate: 45.3% Some college/technical school: 41.9%	HPV
Goldberg (2014) [29]	Nigeria	To examine the influence of social networks and social norms in mothers/caregivers immunization decisions and behaviours.	Mothers living in the Health and Demographic Surveillance System in Bungudu	550	100%	25–34: 42.4%	Qu’ranic school: 93.3%	All routine childhood vaccinations, e.g., Hepatitis B, Measles
Hernandez, Pullen & Brauer (2019) [30]	USA	To examine the role of social networks in decision making of H1N1 vaccination decisions during pregnancy.	Pregnant with first child	223	100%	29.9 (5.3)	Bachelor’s degree: 38.8%	H1N1
Mascia et al. (2020) [31]	Italy	To explore the relationship between students’ vaccination behaviour and their friendship social networks.	Children up to 12 years	49	45.0%	NR	Children in Class 1: 37%	All routine childhood vaccinations, e.g., Hepatitis B
Nyhan et al. (2012) [32]	USA	To examine the effects of social networks on perceptions and vaccination behaviour.	Undergraduate university students	1018	64.0%	NR	Undergraduate university students: 100%	H1N1, seasonal flu
Ruiz (2015) [33]	USA	To assess HPV vaccination sources of information, knowledge, adoption and social networks among young adults.	Undergraduate university students	346	66.2%	20.22 (3.5)	Senior students: 40% Junior students: 39%	HPV

Note. HPV = human papillomavirus; NR = Not reported. ^1^ All studies used a cross-sectional research design. ^2^ When the mean was not reported, the median or the percentage of participants in the age category with most people was reported instead. ^3^ The percentage of participants in the category with most people was reported.

**Table 2 vaccines-09-00607-t002:** Results of studies on the influence of social network members on individuals’ vaccination attitudes and uptake.

Study	Analytical Approach	Social Contagion Results			Impact of Social Networks on Vaccinations	Other Findings
		Clustering ^1^	Centrality ^2^	Homophily ^3^		
Childhood vaccinations (*n* = 4)						
Brunson (2013) [23]	SNA: - 3 models examining influence of beliefs on vaccination: (a) parent (b) people network (c) source network	NR	NR	NR	Non-vaccination increased when having more non-conformers ^4^ in network (OR = 30.57, CI: 5.75–162.65).	Non-conformers ^4^ were more likely to have higher education (i.e., graduate degree; OR = 5.34, CI: 1.05–27.08)
Fu et al. (2019) [28]	LR: - MLS to examine association of parental trust in social contacts for vaccinations and exposure to anti- and pro-HPV vaccine viewpoints ^5^	NR	NR	Participants tended to have similar social networks to themselves: - Mostly female - African American - Parents	Higher HPV refusal was associated with high exposure to anti-vaccine viewpoints (AOR = 1.5, 95% CI: 1.01–2.3) and low exposure to pro-vaccine viewpoints ^5^ (AOR = 1.7, 95% CI = 1.2–2.6). 62.5% of participants holding negative vaccination attitudes reported family and friends having negative vaccination beliefs.	The vaccine advice networks were small, dense, family centric, and homophilous.
Goldberg (2014) [29]	SNA: - LR and MLS models using logit and xtlogit functions	NR	Centrality did not predict vaccination uptake	Participants tend to have similar peers in networks: - Married - Same ethnicity (Hausa, Muslim) - Having no formal education - Similar in co-wife and wealth status	Greater participants’ decision on vaccinating their children was related to the descriptive norm ^6^ (b = 0.92, CI: 0.04–1.7, *p* = 0.04) and injuctive norm ^6^ (b = 2.3, CI: 0.00–0.31, *p* = 0.05) of peers. Both norms of opinion leaders^7^ were not related to participants’ decision on vaccinating their children (*p* > 0.05).	- Frequency of communication with opinion leaders (b = 2.7, CI: 0.58–3.0, *p* = 0.04) and peers (b = 0.63, CI: 0.35–1.6, *p* = 0.02) strengthened the influence of descriptive norms ^6^. - Injuctive norms ^6^ in peer networks were more influential than descriptive norms.
Mascia et al. (2020) [31]	SNA: - MRQA procedures to explore factors associated with formation of network ties and adoption of similar behaviour - LRQA procedure to produce estimates of regression models	NR	NR	Vaccination uptake was more similar in students with the same ethnicity (OR = 5.39–6.13), different gender (OR = 0.84–0.87) and belonging to the same class (OR = 1.68–1.82).	Students were more likely to report similar vaccination uptake with friendship ties occurring after school rather than those established during school (OR = 1.47).	-
Self-vaccination (*n* = 7)						
Casillas et al. (2011) [24]	LR: - 2 MLS models examining the relationship between (a) Source of information model and (b) Discussion about vaccination, on perceived HPV vaccine effectiveness	NR	NR	NR	Participants were more likely to perceive the vaccine as effective: - When hearing about vaccination from family, friends or doctor/nurse/healthcare provider (OR = 4.78, 95% CI: 1.76–12.98). - When discussing (once or more) vaccination with family and/or friends (OR = 1.98, 95% CI: 1.04–3.78).	Having high school education as the highest education level decreased the odds of perceived vaccine effectiveness compared to no school and college levels (OR = 0.47, 95% CI: 0.23–0.96)
Edge et al. (2015) [25]	SNA: - Assortativity coefficient ^8^ to test clusters. - Each individual’s influence on network measured in terms of how well connected they were within network, with between-ness score.	No clustering observed between vaccinated and non-vaccinated individuals	NR	NR	Participants were more likely to get vaccinated if they perceived their peers as being vaccinated (no statistical information reported).	-
Edge et al. (2019) [26]	SNA: - Assortativity coefficient ^9^ for homophily - Auto-logistic regression model: effect of an individual’s social connections on their vaccination decision.	NR	NR	No homophily observed (Assortativity = −0.03, 95% CI: −0.12–0.10)	Participants were more likely to get vaccinated if they had a higher proportion of vaccinated neighbors in their social network (OR = 2.63, 95% CI: 1.28 −5.38).	-
Frank (2011) [27]	SNA: - Primary measure: node’s ^9^ degree of connection with other nodes - HLM and HGLM to examine group influences on health-related attitudes and behaviours	People in the same working group in the company	NR	NR	Participants were more likely to get vaccinated when they perceived their group members as vaccination supporters (γ = 0.08, t = 2.7, *p* < 0.01). People with children were more likely to intend to self-vaccinate (γ = 1.14, t = 2.03, *p* < 0.05). Subjective norms (γ = 0.05, *p* < 0.05) and descriptive norms ^10^ (γ = 0.03, *p* < 0.05) were positively associated with vaccination intention.	-
Hernandez, Pullen and Brauer (2019) [30]	SNA: - Bayesian structural equation modelling	NR	NR	Well-educated women tend to have well-educated networks who support vaccination uptake	Participants were more likely to be vaccinated if they had more network members who were both college-educated and either vaccine supporters (b = 0.35, 95% CI: 0.03–0.66, *p* = 0.01), or discussants (b = 0.10, 95% CI: 0.00–0.27, *p* = 0.02). Participants were less likely to be vaccinated if their network was less educated (none being college-educated) or supporting less vaccination.	-
Nyhan et al. (2012) [32]	LR: - OLS with AOR reported	NR	NR	NR	Participants with more pro-vaccination ^5^ discussion networks reported higher beliefs in vaccine safety (AOR = 1.85–2.32, 95% CI: 1.57–2.84) and greater vaccination intention (AOR = 1.74–1.78, 95% CI: 1.47–2.16). Participants who perceived parents, spouses, or friends as being pro-vaccinated were more likely to report that vaccines are safe (AOR = 1.96–5.59, 95% CI: 1.25–12.57) and greater vaccination intention (AOR = 1.52–2.49, 95% CI: 0.66–5.56).	-
Ruiz (2015) [33]	LR: - BLS to test relationship between network density ^11^ and homophily on vaccine adoption status.	NR	NR	NR	Higher vaccination uptake, compared to non-vaccination, was associated with: - Perceptions that family members were vaccinated (B(1) = 2.41, *p* < 0.05) - Made themselves the decision to be vaccinated (B(1) = 0.89, *p* < 0.05) - Their parents were part of vaccination decision-making (B(1) = 1.61, *p* < 0.05) - Lower density ^11^ in social networks (B(1) = 0.30, *p* < 0.05).	Vaccinated participants were more likely to trust family members (75%) for information about vaccines compared to non-vaccinated (60%) (*p* < 0.05).

Note. AOR = Adjusted Odds Ratio; BLS = Binomial logistic regression; CI = Confidence Interval; HGLM = hierarchical generalized linear modelling; HLM = hierarchical linear modelling; LR = Logistic Regression; LRQA = Logistic regression quadratic assignment; MLS = Multivariate logistic regression; MRQA = Multiple regression quadratic assignment; NR = Not reported; OLS = Ordered logistic regression; OR = Odds Ratio; SNA = Social network analysis. ^1^ Clustering: co-occurrence of a trait in connected individuals. ^2^ Centrality: the position of a node within a network. ^3^ Homophily: the tendency to relate to people with similar characteristics. ^4^ Conformers: Parents who conform to the nationally recommended vaccination schedule by having their children vaccinated completely and on time; Nonconformers: parents who did not conform to the nationally recommended vaccination schedule by delaying vaccination, partially vaccinating, or not vaccinating at all. ^5^ Anti-vaccine viewpoints: negative viewpoints on vaccinations; Pro-vaccine viewpoints: positive viewpoints on vaccinations. ^6^ Descriptive norm: Observing peers/opinion leaders immunizing their own child. Injunctive norm: Perceiving that the majority of peers/opinion leaders supporting immunizations. ^7^ Opinion leaders: religious leaders, political leaders, and traditional medicine providers. ^8^ Assortativity coefficient is a standard network measure developed by Newman (2002) to examine clustering or homophily in a specific population. ^9^ Node: the people comprising a social network (e.g., study participants). ^10^ Subjective norms: those who felt that relevant others wanted them to get the vaccination and who felt motivated to comply with those relevant others; Descriptive norms: the percentage of people that respondents think engage in the specified behaviours. ^11^ Density: a measure of how well connected a network is and is often used to compare networks against each other.

**Table 3 vaccines-09-00607-t003:** Needs for further research based on types of social network, vaccinations and attitudes.

Type of Social Network	Vaccination Type	Studies	Vaccination Attitude	Further Research
General	All self and childhood	-	Positive Negative	- Examine influence of social networks on vaccination attitudes and uptake of individuals longitudinally, using sociocentric networks - Examine position within a network (centrality) and whether it is associated with greater/lower vaccination uptake - Examine whether clustering exists with specific members of social network and how it influences vaccination attitudes and uptake of individuals - Examine if homophily exists longitudinally
Family/spouses/partners	H1N1 (Self) HPV (Self and Childhood) Seasonal flu (Self)	Casillas et al. (2011) [24] Fu et al. (2019) [28] Hernandez et al. (2019) [30] Nyhan et al. (2012) [32] Ruiz (2015) [33]	Positive	Mechanisms underlying why: - Family, peers and friends have higher influence on vaccination attitudes and uptake of individuals than other members in network including healthcare professionals and politicians - Perceiving family, peers and friends as vaccination supporters/hesitant is associated with greater/lower vaccination uptake in individuals - Having a greater number of family, peers and friends in social networks who are vaccinated/under-vaccinated influence similarly vaccination uptake - Observing friends/peers vaccinating their child influence vaccination uptake of individuals on their own child (e.g., imitation behaviour)
	All routine childhood H1N1 (Self) HPV (Childhood) Seasonal flu (Self)	Brunson et al. (2013) [23] Fu et al. (2019) [28] Hernandez et al. (2019) [30] Nyhan et al. (2012) [32] Ruiz (2015) [33]	Negative	
Friends/Peers	All routine childhood H1N1 (Self) HPV (Self and Childhood) Seasonal flu (Self)	Casillas et al. (2011) [24] Edge et al. (2015) [25] Fu et al. (2019) [28] Goldberg (2014) [29] Hernandez et al. (2019) [30] Mascia et al. (2020) [31] Nyhan et al. (2012) [32]	Positive	
	All routine childhood H1N1 (Self) HPV (Childhood) Seasonal flu (Self)	Brunson et al. (2013) [23] Fu et al. (2019) [28] Goldberg (2014) [29] Hernandez et al. (2019) [30] Nyhan et al. (2012) [32]	Negative	
Health Care Providers	All routine childhood H1N1 (Self) HPV (Self) Seasonal flu (Self)	Casillas et al. (2011) [24] Goldberg (2014) [29] Hernandez et al. (2019) [30] Nyhan et al. (2012) [32]	Positive	- Mechanisms underlying why healthcare providers have less influence on vaccination attitudes and uptake than other network members including family and friends - Possible factors to be explored: sociodemographics, non-central position in the social network, quality of the relationship with the individual and frequency of communication
	All routine childhood H1N1 (Self) Seasonal flu (Self)	Brunson et al. (2013) [23] Goldberg (2014) [29] Hernandez et al. (2019) [30] Nyhan et al. (2012) [32]	Negative	
Co-workers	H1N1 (Self)	Frank (2011) [27]	Positive	Mechanisms underlying why: - Perceiving co-workers as vaccination supporters/hesitant is associated with greater/lower vaccination uptake in individuals - Having a greater number of co-workers who are vaccinated/under-vaccinated influence similarly vaccination uptake - Examine the influence of co-workers on attitudes and uptake of HPV and seasonal flu vaccinations - Compare influence of co-workers with other network members including family and friends - Examine specific characteristics associated with clustering observed such as position in work
	All routine childhood H1N1 (Self)	Brunson et al. (2013) [23] Frank (2011) [27]	Negative	
Politicians	All routine childhood	Goldberg (2014) [29]	Positive	- Examine whether specific vaccination behaviours (e.g., observing them being vaccinated) influence individuals’ vaccination uptake - Examine whether negative attitudes or lower vaccination uptake of politicians influence in the same way individuals - Compare influence of politicians with other network members including family and friends - Examine the influence of politicians on attitudes and uptake of HPV, H1N1 and seasonal flu vaccinations
	-	-	Negative	
Neighbours	Seasonal flu (Self)	Edge et al. (2019) [26]	Positive	- Examine whether vaccination attitudes and uptake of neighbours living in smaller and bigger cities as well as in general population influences in the same way those of individuals - Compare influence of neighbours with other network members including family and friends - Examine the influence of neighbours on attitudes and uptake of HPV, H1N1 and childhood vaccinations
	-	-	Negative	

Note. HPV = Human Papillomavirus.

## Data Availability

The data supporting the publication are available through Open Science Framework (OSF) (osf.io/5 gucf).

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
