# Peer review of "Transmission of Vaccination Attitudes and Uptake Based on Social Contagion Theory: A Scoping Review"

_vaccines, 2021, doi:10.3390/vaccines9060607_

Round 1
Reviewer 1 Report
The manuscript titled “Transmission of vaccination attitudes and uptake based on Social Contagion Theory: A scoping review” proposed a review on primary studies investigating how attitudes and vaccination uptake of individuals might be related to and influenced by their social network. A final sample of 11 primary studies was included in the review, and a narrative synthesis approach was employed to analyze findings. Results showed that family and friends/peers had a major influence – in both positive and negative directions - on self and childhood vaccination attitudes and uptake compared to other network members (i.e., co-workers, neighbours, politicians, etc.) Another relevant factor found to influence the transmission of vaccination attitudes and uptake was homophily. Authors highlighted that social networks play a key role in shaping attitudes and vaccination uptake, discussed their results in light of previous literature and proposed hint for further research.
I carefully read the manuscript, and I think it may be of interest for the readers of Vaccines. The manuscript is very well-written and properly addresses the interesting issue of the relationship between social network membership and vaccination attitudes and uptake. The topic is really relevant nowadays, and more knowledge coming from secondary studies is needed. I also appreciated the construction of a model in order to summarize the main findings. I found that the introduction section and the methodology employed are clear and detailed, as well as the explanations provided in the discussion section. I only have few minor remarks:
Introduction Section
Page 2, lines 89-91: The two aims presented seem to refer to a primary study. My suggestion is to expand the aims and to reformulate them so as to refer to a secondary study (e.g., “to describe the features of social network membership within the included studies”).
Materials and Methods Section
Page 3 lines 119-122: I do not understand where the two indices (IRR and k) come from. You stated that inter-rater reliability was calculated using Cohen's kappa. So Cohen's kappa is the index used for estimating inter-rater reliability. What about IRR, what does it represent and how was it calculated?
Results
Page 4 lines 154-155: Please, add information about the analyses employed in the remaining 4 studies. Which kind of analysis did they use to obtain results?
Page 12 lines 200-203, note of Figure 1: Please explain how you concluded that there was a lower rather than a higher influence of social networks on vaccination attitudes across studies.
Page 12 lines 204-205: Similarly to my previous comment, how can you say that "family and friends/peers had the highest effect in..." How was that effect established?
Author Response
Response to Reviewer 1 Comments
Point 1: The manuscript titled “Transmission of vaccination attitudes and uptake based on Social Contagion Theory: A scoping review” proposed a review on primary studies investigating how attitudes and vaccination uptake of individuals might be related to and influenced by their social network. A final sample of 11 primary studies was included in the review, and a narrative synthesis approach was employed to analyze findings. Results showed that family and friends/peers had a major influence – in both positive and negative directions - on self and childhood vaccination attitudes and uptake compared to other network members (i.e., co-workers, neighbours, politicians, etc.) Another relevant factor found to influence the transmission of vaccination attitudes and uptake was homophily. Authors highlighted that social networks play a key role in shaping attitudes and vaccination uptake, discussed their results in light of previous literature and proposed hint for further research.
I carefully read the manuscript, and I think it may be of interest for the readers of Vaccines. The manuscript is very well-written and properly addresses the interesting issue of the relationship between social network membership and vaccination attitudes and uptake. The topic is really relevant nowadays, and more knowledge coming from secondary studies is needed. I also appreciated the construction of a model in order to summarize the main findings. I found that the introduction section and the methodology employed are clear and detailed, as well as the explanations provided in the discussion section. I only have few minor remarks:
Response 1: We would first like to thank the reviewer for recognizing the need of this review, for the comments and for allowing us to edit the manuscript so to make a stronger effort to advance current knowledge in the field. We have added all relevant information in the paper as suggested, hoping that we have addressed sufficiently the reviewer’s comments.
Introduction Section
Point 2: Page 2, lines 89-91: The two aims presented seem to refer to a primary study. My suggestion is to expand the aims and to reformulate them so as to refer to a secondary study (e.g., “to describe the features of social network membership within the included studies”).
Response 2: As recommended, the two aims were changed in order to refer to a secondary study:
“The main objectives are: a) to describe the features of social network membership within the included studies, and b) to examine the evidence from the included studies on how vaccination attitudes and uptake of individuals are influenced by their social networks.”
Materials and Methods Section
Point 3: Page 3 lines 119-122: I do not understand where the two indices (IRR and k) come from. You stated that inter-rater reliability was calculated using Cohen's kappa. So Cohen's kappa is the index used for estimating inter-rater reliability. What about IRR, what does it represent and how was it calculated?
Response 3: Inter-rater reliability (IRR) is the extent to which two raters agree. In our study, it was assessed using the percent agreement (number of agreement scores divided by the total number of scores) and Cohen’s kappa (a more robust measure for IRR). We used both indices in order to provide readers with more information on the robustness of the evidence we provide. For better clarification, we have added this information in Methods section “2.2. Search strategy and study selection”, page 3:
“Inter-rater reliability (IRR) referring to the extent to which the two screeners agreed, was calculated using the percent agreement (number of agreement scores divided by the total number of scores) and Cohen’s kappa (a more robust measure for IRR) [20]. An almost perfect agreement was observed between the two screeners during title/abstract screening (IRR=94%; k=.86) and for the full-text screening stage (IRR=96%; k=.86).”
Results
Point 4: Page 4 lines 154-155: Please, add information about the analyses employed in the remaining 4 studies. Which kind of analysis did they use to obtain results?
Response 4: As suggested, information on the analytical approach implemented by the remaining four studies was added in page 4:
“Most of the included studies used social network analysis to examine influence of social networks on vaccination attitudes and uptake (n=7, 63.6%) and with the remaining using logistic regression models (n=4, 36.4%).”
Point 5: Page 12 lines 200-203, note of Figure 1: Please explain how you concluded that there was a lower rather than a higher influence of social networks on vaccination attitudes across studies.
Response 5: As recommended, explanation on how we have concluded that higher and lower influence of each social network members on vaccination attitudes and uptake of individuals was added, in page 11, note of Figure 2. Further, a specification in the caption of the Figure 2 was added to better clarify this.
Point 6: Page 12 lines 204-205: Similarly to my previous comment, how can you say that "family and friends/peers had the highest effect in..." How was that effect established?
Response 6: We thank the reviewer for noticing this and we have now clarified the reference of 'higher effect' which we have observed in the included studies. We came into these conclusions based on the total number of studies providing Odds Ratios which compare social network members. Specifically, eight studies (73%) reported that family and friends/peers significantly influenced self and childhood vaccination attitudes and uptake. In contrast, only two studies (18%) reported that healthcare providers and co-workers and one study (9%) that politicians [29] significantly influenced vaccination attitudes and uptake. To better clarify this, this information was added in page 11:
“Most of the included studies (n=8, 72.7%) reported that family and friends/peers significantly influenced self and childhood vaccination attitudes and uptake. In contrast, only two studies (18.2%) reported that healthcare providers [29,32] and co-workers [23,27] and one study (9.0%) that politicians [29] significantly influenced vaccination attitudes and uptake. For example, Casillas et al. [24] reported that discussing about the vaccine with family and/or friends significantly increased the odds for perceiving the HPV vaccine as effective (Odds Ratio=1.98, 95% CI: 1.04-3.78) compared to discussing them with the healthcare provider which had a non-significant effect (Odds Ratio=1.71, 95% CI: .86-3.39).”

Reviewer 2 Report
- Please rephrase line 16.
- Some terminologies need to be explained before quoting them in abstract.
- For ease, please mention as hesitant to take the vaccine in line 25.
- The manuscript has to be checked for grammar discrepancies.
- Fig 1 & 2 are quite blurred please use a better ones
- Factors that you had included in the study as age, education, %age female, need to be discussed as well, if no co-relation was found or further studies are needed please specify in the limitation section.
Author Response
Response to Reviewer 2 Comments
Point 1: Please rephrase line 16.
Response 1: As suggested, line 16 was rephrased:
“Vaccine hesitancy is a complex health problem, with various factors involved including the influence of an individual’s network.”
Point 2: Some terminologies need to be explained before quoting them in abstract.
Response 2: We have carefully read the abstract and identified that the terminology of homophily was reported and needed further explanation (please note that we attempted to add this while trying to remain brief due to abstract word limitations):
“Homophily -the tendency of similar individuals to be connected in a social network- was identified…”
Point 3: For ease, please mention as hesitant to take the vaccine in line 25.
Response 3: As suggested, “vaccine hesitant” was changed into “hesitant to take the vaccine”.
Point 4: The manuscript has to be checked for grammar discrepancies.
Response 4: The paper was carefully proofread and corrections were made throughout.
Point 5: Fig 1 & 2 are quite blurred please use a better ones
Response 5: We apologize for this. As suggested, we have replaced Figures 1 and 2 with ones with better resolution.
Point 6: Factors that you had included in the study as age, education, %age female, need to be discussed as well, if no co-relation was found or further studies are needed please specify in the limitation section.
Response 6: Thank you for the comment. We note that some of these factors were examined in included studies and some were not. Specifically, educational level, parental and marital status were found as impacting the formation of social networks with members tending to have similar networks in respect to these characteristics. As suggested, we have added in the first paragraph of page 16:
“Substantial homophily was identified in included studies, with race/ethnicity playing the most important role in forming social networks. Other factors identified being similar between network members were education level, parental and marital status […] For example, in parental populations, public health vaccination campaigns could emphasize the protection of their children from health conditions, whereas in non-parental populations could emphasize the protection of themselves and significant others. Targeting each network using recommendations for campaign messaging such as the use of short, risk-reducing or relative risk framing messages with clear and simple language, [45] could possibly reduce vaccine hesitancy.”
Further, mixed findings were observed for gender, with some people tending to have mostly people with the same gender in their networks and others people with the opposite. In regards to age, it was not examined in any of the studies. Thus, we have added in the second paragraph of page 16:
“Additionally, future studies are suggested to examine the impact of specific sociodemographic characteristics such as age and gender in forming social networks and in the spread of vaccination attitudes and uptake as either were not examined in the included studies or mixed findings were observed (i.e., same vs. opposite gender) [28,31].”

Reviewer 3 Report
In the manuscript titled “Transmission of vaccination attitudes and uptake based on Social Contagion Theory: A scoping review” by Kassianos et.al., the authors gave a very comprehensive review of how the attitude and awareness of people are affecting the recent vaccine hesitancy. The review describes how attitudes and vaccination uptake are spread within the social network based on the Social Contagion Theory. The authors did a very rational examination of various databases. In spite of having different limitations, the authors showed how one’s social network affects the vaccination attitude and uptake. This reviewer believes this manuscript is good for acceptance in Vaccines after modifying a minor concern.
The authors are recommended to revise the figures (Figure 1 and 2). They are not clear.
Author Response
Response to Reviewer 3 Comments
Point 1: In the manuscript titled “Transmission of vaccination attitudes and uptake based on Social Contagion Theory: A scoping review” by Kassianos et.al., the authors gave a very comprehensive review of how the attitude and awareness of people are affecting the recent vaccine hesitancy. The review describes how attitudes and vaccination uptake are spread within the social network based on the Social Contagion Theory. The authors did a very rational examination of various databases. In spite of having different limitations, the authors showed how one’s social network affects the vaccination attitude and uptake. This reviewer believes this manuscript is good for acceptance in Vaccines after modifying a minor concern.
Response 1: We would like to thank the reviewer for recognizing the great deal of work which we have put in this manuscript and the need of this review. We hope the added information addressed the reviewer’s concern.
Point 2: The authors are recommended to revise the figures (Figure 1 and 2). They are not clear.
Response 2: Thank you for the comment. We were unsure whether the reviewer was referring to the quality of the figures or the text. Therefore, based also on the other reviewers’ comments we made changes to both. Specifically, we have replaced Figures 1 and 2 with ones with better resolution. Also, we have provided more information on the caption of Figure 1 and on the notes of Figure 2.
